# The Influence of Depression on Biased Diagnosis of Premenstrual Syndrome and Premenstrual Dysphoric Disorder by the PSST Inventory

**DOI:** 10.3390/life11111278

**Published:** 2021-11-22

**Authors:** Andrzej Śliwerski, Karolina Koszałkowska

**Affiliations:** Institute of Psychology, University of Łódź, 90-136 Łódź, Poland; karolina.koszalkowska@uni.lodz.pl

**Keywords:** premenstrual syndrome, premenstrual dysphoric disorder, item response theory analysis, depression

## Abstract

The diagnosis of premenstrual syndrome (PMS) and premenstrual dysphoric disorder (PMDD) poses a challenge for clinicians due to the overdiagnosis of retrospective methods and overlapping symptoms with depression. The present study utilized an Item Response Theory analysis to examine the predictive utility of the Premenstrual Symptom Screening Tool (PSST) in women with and without depression. Two hundred and fifteen women aged 20–35 completed the PSST, a daily symptom calendar, SCID-I, and CES-D for two consecutive menstrual cycles. PSST items: fatigue, depressed mood, feeling overwhelmed, anxiety/tension, and decreased interest in everyday activities were the best predictors of PMS. Unlike the daily symptom ratings, the PSST over-diagnosed PMS/PMDD in the depressed group but not in the group of women without PMS/PMDD. While diagnosing premenstrual disorders, clinicians should be aware that a retrospective diagnosis with PSST can be more sensitive to mood disorders and cycle phases than a prospective diagnosis with a daily symptoms calendar.

## 1. Introduction

The term premenstrual syndrome (PMS) refers to a group of recurrent physical and psychological symptoms that occur during the luteal phase of the menstrual cycle and subside at the beginning of or during the menstrual period, affecting between 30% and 80% of reproductive-aged women [1,2]. The core symptoms of PMS identified in prospective daily ratings in a large sample of women included anxiety/tension, mood swings, aches, food cravings, cramps, and decreased interest in activities [3]. The presence of emotional symptoms is not crucial for the diagnosis of PMS, which can be solely based on physical symptomatology if it is burdensome enough to cause functional impairment [4]. Although many common physical, behavioral, and emotional symptoms have been recognized through empirical observations made from women’s daily rating of symptoms, there is still no unified and standardized list of criteria for PMS [1].

Premenstrual Dysphoric Disorder (PMDD) is often described as a severe form of PMS in which affective symptomatology is pronounced, leading to a significant interference with women’s quality of life [5]. PMDD is recognized as a separate category of affective disorders in the 5th edition of the Diagnostic and Statistical Manual of Mental Disorders [6] and, thus, has become much more standardized and precisely defined than PMS [7]. PMDD is characterized by marked affective, cognitive, and physical symptoms that occur repeatedly before menstruation and remit at the onset of, or shortly after, menses. PMDD is differentiated from PMS by the severity of the symptoms, as well as the requirement that they cause significant distress or impairment, interfering with one’s activity at work, school, or social/family life [6]. It is estimated that PMDD affects around 3–5% of women of reproductive age [8], but a diagnosis made by strict application of the DSM criteria decreases that number to 1 to 2% [6,9].

Many different assessment tools can be used to determine the extent to which women suffer from premenstrual symptoms, including both prospective, self-administered calendars and retrospective screening tools used by clinicians [5]. Among the most commonly used are the Calendar of Premenstrual Experiences [10], Daily Record of Severity of Problems [11], and Premenstrual Symptoms Screening Tool [12].

Despite the fact that both PMS and PMDD have long been recognized in the medical and psychological literature, their diagnosis poses a challenge for clinicians and health professionals for various reasons. Notably, many women who experience clinically significant premenstrual symptoms do not meet the full diagnostic criteria of PMDD and end up underrecognized [13,14]. Moreover, issues with the under- and overestimation of premenstrual disorders may also depend on the diagnostic tools used in the diagnostic process. Comparisons between a prospective, self-administered questionnaire (DRSP) and a retrospective screening tool (PSST) revealed that the PSST overestimated the diagnosis of PMDD, while the diagnosis of PMS was underreported. Moreover, the diagnosis of PMS was higher in the prospective, diary-like tool (DRSP) [15]. It is generally agreed that prevalence estimates of PMDD tend to inflate upon retrospective rather than prospective methods [6], most likely because of the recall bias related to one’s memory [4,16]. Importantly, different tools vary greatly in terms of the criteria used in their assessment of premenstrual symptoms. There is little consensus regarding the number of days considered as part of a particular cycle phase or the assessment of functional impairment in the case of PMDD diaries [17]. Taken together, it seems that there is a great deal of uncertainty when dealing with premenstrual symptoms, especially in the way one should differentiate between PMS and PMDD in terms of their core symptomatology. This uncertainty tends to depend heavily on the assessment tools being used, as they prioritize PMS/PMDD symptomatology in a diverse manner.

Another challenge in recognizing PMS and PMDD is the extent to which symptoms of premenstrual and depressive disorders overlap and interrelate with each other. A large population-based study revealed that the prevalence of major depression (MD) was 11.3% in women with moderate PMS and 24.6% in women with severe PMS [18]. A study of premenopausal women revealed that 57.6% of those meeting the criteria for PMDD also had a history of MD, while, in the non-PMDD group, only 28.1% were previously diagnosed with MD [19]. Kepple and colleagues [20] found that mood disorders, primarily MD, are more likely to cooccur with PMDD (43.2%) than other Axis I conditions, such as anxiety disorders (15.3%). Another study reported comorbidity rates for PMDD and mental disorders reaching 47.7% for anxiety disorders, 22.9% for mood disorders, and 28.4% for somatoform disorders [21]. Moreover, pharmacologically induced hormonal changes have been found to trigger mood symptoms specifically in women with a history of depression [22]. In fact, episodes of depression caused by various hormonal fluctuations (occurring specifically during the premenstrual, postpartum, and perimenopausal phases in women) have been recognized as the reproductive subtype of depression [23]. Taken together, all of these results demonstrate the relevance of recognizing and controlling for potentially cooccurring affective disorders in the process of diagnosing, treating, and preventing PMS/PMDD.

The main aim of the present study was to verify how depression affects the diagnosis of PMS and PMDD. We hypothesized that depression would affect the overdiagnosis of PMS/PMDD in both prospective and retrospective methods. Another goal of the study was to examine which items of the PSST have the greatest discriminatory value in order to further determine which symptoms diagnose PMS and PMDD most accurately. Due to the fact that some of the symptoms of PMS and PMDD coincide with those of depression, we attempted to verify whether the discrimination power of each item was different between the groups of women with and without depression. Finally, we wanted to verify how the sensitivity and specificity of the PSST and daily calendar would differ for the groups with and without depression. We hypothesized that the PSST items would have different diagnostic powers and that depression would affect the responses in only some of them.

## 2. Materials and Methods

### 2.1. Procedure and Participants

Women aged 20–35 were invited to participate in a study for women with or without premenstrual problems. They were recruited through advertisements on internet portals and social media between June 2018 and February 2019. The invitation for the study contained information about the aim of the study, inclusion criteria, and remuneration for participation in the study. Women were required to have regular cycles lasting between 20 and 34 days. Exclusion criteria were the diagnosis of any hormonal problems (like hyperthyroidism, hypothyroidism, and hyperprolactinemia); endometriosis; amenorrhea; on-going pregnancy; or fertility problems. Women who agreed to participate were asked to fill in an informed consent form for the study, a screening questionnaire to verify their depression symptoms, and self-reporting questions. The women were asked to make daily ratings of their symptoms for a minimum of two consecutive menstrual cycles. After completing the monitoring of the first cycle, they took part in a structured interview to verify the occurrence of depression. Women were randomly assigned to groups in which the PSST was filled either in the follicular or luteal phase. Participation was remunerated upon completion of all stages of the study.

In total, 427 women agreed to participate in the study. After reviewing the study objectives and requirements, 39 refused to participate. One hundred and twenty women did not complete the monitoring of their menstrual cycles, and 53 women were excluded (2 became pregnant, and the rest withdrew their participation without giving any reasons). Finally, the results from 215 women who fulfilled the eligibility criteria and completed all the tasks were included in the study (Figure 1). The vast majority of participants did not have any children (91.6%), 5.58% of them had one child, and 2.33% had two children. Detailed descriptive statistics of the final sample are given in Appendix A. On the basis of the structured interview, 86 women met the criteria for major depression. Some of the study participants used psychological or psychiatric help for a depressive disorder or anxiety. Women from the nondepressed group who participated in psychiatric care/psychotherapy have had anxiety problems.

### 2.2. Instruments

#### 2.2.1. Depression

Four hundred and twenty-seven women were screened by The Center for Epidemiologic Studies—Depression Scale (CES-D). The CES-D includes 20 items for self-reported measurements of symptoms associated with depression experienced in the past week. The Polish version of the CES-D showed excellent internal consistency (a = 0.90) [24]. The CES-D results allowed us to diagnose depressive mood pre-control of how many participants we had in the depressed and nondepressed groups. However, the final division was made by a clinical interview.

Two hundred and fifteen participants were evaluated diagnostically using the Structured Clinical Interview for DSM-IV (SCID-I) [25]. The interviewers were blind to all of the participants’ scores. On the basis of the structural interview, a diagnosis of depression occurring at the time of the study, as well as in the past, was carried out. Women diagnosed with depression and those who previously experienced depression (within the last 3 years) were included in the depression group.

#### 2.2.2. Prospective PMS/PMDD Diagnosis

All participants recorded their PMS symptoms using the Calendar of Premenstrual Symptoms every day for two consecutive menstrual cycles. The tool consisted of 11 items representing various symptoms of premenstrual distress: depressed mood or dysphoria, anxiety/tension, mood lability, irritability, decreased interest in usual activities, poor concentration, marked lack of energy, marked change in appetite, hypersomnia or insomnia, and feeling overwhelmed, as well as physical symptoms such as bloating. Finally, participants rated the extent to which these symptoms interfered with their work, activities, or relationships. Participants rated the daily severity of the symptoms on a 4-point scale. The calendar was administered using a custom-built website. The women were instructed to log into their personal account on the website every day for two consecutive menstrual cycles and rate their symptoms. If the participant forgot to complete the calendar, she received a reminder from the system. The effect of such reminders was that there were no missing data in the completed calendars. Due to the fact that PMS/PMDD symptom severity may vary during the day [17], participants were asked, if possible, to complete their calendars at the same part of the day throughout the whole menstrual cycle. Calendar scores were summed across the daily symptom reports for three five-day periods: last days of the luteal phase, first days of the menstrual cycle, and the follicular phase (from 12–16 days after the last menses). For a PMS diagnosis, participants had to mark at least 5 symptoms as moderate or severe for at least 2 days in the luteal phase without showing any symptoms in the middle of the cycle. Additionally, they had to point out that their symptoms interfered with their functioning at least in a moderate way. The same criteria were used for a PMDD diagnosis, except that the symptoms had to be specified by participants as severe.

#### 2.2.3. Retrospective PMS/PMDD Diagnosis

The participants filled out the Premenstrual Symptoms Screening Tool (PSST) [12] in either the follicular or luteal phase of the cycle. As the PSST was developed according to DSM diagnostic criteria, its items were divided into two parts: one part evaluating the intensity of 14 symptoms in the luteal phase of the cycle and the other assessing the degree to which these symptoms influenced one’s effectiveness at work, activity, and relations with people. Each item was rated on a four-point scale (not at all, mild, moderate, and severe). The PMS group was differentiated from the non-PMS group, according to the DSM-5 criteria: at least 5 moderate or severe symptoms from the first part, with at least one being from the first four symptoms. Moreover, in the PMS group, it was required that that the symptoms interfered with one’s functioning at least moderately. The same criteria were used for a PMDD diagnosis, except that the symptoms and list from the second part had to be specified by participants as severe.

### 2.3. Data Analysis

To verify the hypothesis that depression affects the diagnosis of PMS/PMDD made through prospective and retrospective methods, we utilized both the Item Response Theory (IRT) and sensitivity/specificity calculations. IRT models help explain the relationship between latent traits (symbolized by *θ*) and their manifestation in questionnaire responses. It is assumed that the items of a scale are not equally informative across the latent trait range. An item’s informative value depends on its two parameters: difficulty and discrimination. The difficulty parameter (b) is indexed by how much of the trait is needed to answer the item correctly. In other words, the more difficult the item is, the less likely a person with a low value of the trait will answer in accordance with the key. The item discrimination parameter (a) describes how well the item differentiates among individuals at a specific level of the trait. In such terms, every item provides a different level of information at each level of the trait [26]. Items scored in multiple-ordered categories are referred to as polytomously scored items. In polytomous IRT models, the item characteristic curves (ICC) can be plotted for each response category [27].

To conduct Item Response Theory analyses, the *ltm*, *msm*, and *polycor* packages [28] were used. Due to the Likert-type scale used in the PSST, we used a Graded Response Model (GRM) for the polytomous data [28]. In this model, the difficulty parameters indicate the threshold of latent traits between answer choices, and the discrimination parameter reflects the degree to which an item discriminates individuals across the latent trait range [29]. The *ltm* package provides a flexible framework for IRT analyses for polytomous data under the Marginal Maximum Likelihood approach. To distinguish between the differences in item functioning between women with and without depression, we performed an ordinal logistic regression differential item functioning (DIF) analysis using IRT theta estimates as the conditioning values. For this purpose, we utilized the *lordif* package [30].

Finally, we used the *epiR* package for R [31] to calculate the PSST sensitivity and specificity. A data analysis was conducted with *RStudio* [32], with a *p*-value of <0.05 considered significant. All graphs in this paper were prepared using the *ggplot* package for R.

## 3. Results

During the first step, we checked whether women with depression differ from the nondepressed group in terms of descriptive statistics. The differences in the mean cycle length (t_(213)_ = 0.101; *p* > 0.05), mean age (t_(213)_ = 1.501; *p* > 0.05), number of children (t_(213)_ = 0.302; *p* > 0.05), and taking oral contraceptive (OC) (*χ*^2^(1) = 1.151; *p* > 0.05) did not appear to be meaningful. At the same time, there were no significant differences in the occurrence of PMS/PMDD between women taking OC and those not taking OC (*χ*^2^(1) = 0.142; *p* > 0.05). Moreover, the occurrence of PMS and PMDD diagnosed on the basis of menstrual calendars did not differ between the groups with and without depression (*χ*^2^(1) = 2.502; *p* > 0.05). We also verified whether the phase of the cycle in which the study was conducted differentiated the study variables. There was no difference in depression (*χ*^2^(1) = 0.534; *p* > 0.05) and a prospective diagnosis of PMS/PMDD (*χ*^2^(2) = 0.884; *p* > 0.05). However, the outcome of a retrospective diagnosis with PSST differed depending on the phase of the cycle. More women had PMS when tested in the luteal phase of the cycle (*χ*^2^(1) = 4.188; *p* < 0.05).

### 3.1. Item Response Theory Analysis

In the first step, we evaluated the fits of an unconstrained and a constrained model (assuming nonequal and equal discrimination parameters across items, respectively). The likelihood ratio test for the unconstrained model (Akaike Information Criterion (AIC) = 9113.10; Bayesian Information Criterion (BIC) = 9369.27) was significantly better than for the constrained model (AIC = 9188.92; BIC = 9384.42). Models with lower AIC values are desirable, as they indicate a closer fit to a true model [33].

In the IRT analysis, the discrimination level (a) is considered high if its value is greater than 1. Only two items (10 and 11) reached the low discrimination level. Item 10 (“Overeating/food cravings”) had a discrimination parameter very close to 1 (a = 0.988) in the general sample and was very good (a = 1.784) for the nondepressed sample. However, item 11 had a very low discrimination level in all the samples, which means that insomnia poorly differentiates between women with and without PMS/PMDD. Nine items had α levels greater than 2, which means that these questions were very good at differentiating women with and without PMS/PMDD. The discriminatory values of individual items decreased in the group of women with depression and were very high in the group of women without depression (Table 1). Nevertheless, item 11 had an invariably low discriminatory value across all the groups.

The trace curves for each item were visually inspected to determine the probability levels of giving each answer in the questionnaire depending on the latent trait level. In Figure 2, there are nineteen plots, each for one item.

A trace curves analysis showed that, for items 1 (“Anger/irritability”), 3 (“Tearful”), and 10 (“Overeating/food cravings”), the participants responded in a very dichotomous way, choosing options 1 or 4 significantly more often than the other options. The mild and moderate answers (choice 2 and 3) were less frequent compared to the other item choices, indicating that the individuals were highly confident whether they did or did not have these symptoms at all. For item 11, the probability of a response was independent of the level of the latent trait. The trace curve for this item showed that the majority of women indicated that they did not experience insomnia at all. Items 2 (“Anxiety/tension”), 7 (“Decreased interest in social activities”), and 14 (“Physical symptoms”), as well as items C (“Interference relationships with family”), D (“Interference social life”), and E (“Interference home responsibilities”), had very low curves for answer 3. For these items, women declared either a complete lack of symptoms or their mild/most severe form.

The examination of the trace curve also included a comparison between women with and without depression. To compare the differences in item functioning between women with and without depression, the DIF function was utilized. We used the likelihood ratio (LR) χ^2^ test as the detection criterion at the α level of 0.05 and McFadden’s pseudo R^2^ as the magnitude measure. Seven items displayed depression-related DIF: 1, 2, 3, 8, 11, 12, and 14. In Figure 2, there are additional gray line curves drawn for the items that represent women with depression. On average, depressed women had higher mean scores of anger/irritation, anxiety/tension, and increased sensitivity. The curves for these items were skewed sharply to the left, which indicates that, for the lower levels of PMS, women reported the occurrence of these symptoms. For difficulties with concentration, women from the depressed group were less likely to choose the middle responses. They either declared a lack of disturbance of concentration or its mild or severe form. Differences were also present for the questions concerning sleep disorders. Although a small number of women declared having insomnia, it was slightly more frequent among women with both PMS/PMDD and depression. Hypersomnia, however, was declared much more often by depressed women than by women without depression. The absolute difference between the Item Characteristic Curves (ICCs) for the two groups showed that the item true score functions peaked at approximately *θ* = 0. Thus, the biggest diagnostic error may occur for women with mild PMS symptoms. In their case, one may mistakenly classify symptoms of depression as mild PMS symptoms. At the same time, women without PMS and those with its severe form will be correctly diagnosed regardless of the occurrence of depression.

The inspection of the Item Information Curves showed that item 11 had the lowest discrimination and difficulty rate among the PSST items (Figure 3). Such results indicate that insomnia is a dimension independent of PMS/PMDD. The best discrimination parameters were obtained by fatigue/lack of energy (item 9), depressed mood (item 4), feeling overwhelmed or out of control (item 13), and concentration (item 8), as well as decreased interest in home activities (item 6), in work activities (item 5), and in social activities (item 7). These seven items had the highest informative values, which means that they best differentiated between women with and without PMS/PMDD. This parameter does not, however, override the information value of the remaining items. For example, anger/irritability (item 1) had a good but lower discrimination value, and it also had a low value for the difficulty parameter. This indicates that a large number of women without PMS/PMDD acknowledge that they may experience increased irritation during the luteal phase of their cycle. Furthermore, overeating/food cravings (item 10) and physical symptoms (item 14) were declared by women regardless of the occurrence of PMS.

Next, the Test Information Curve for the PSST was estimated and indicated that the curve covers a desired range of −2 to +2, which is expected of a clinical diagnostic tool. The accuracy of the measurement for women with PMS for the entire scale in the range of −2*θ* to 2*θ* was 76.41 (83.7% of the total information).

### 3.2. Discriminant Validity, Sensitivity, and Specificity

To calculate the sensitivity and specificity of the PSST, the PMS and PMDD diagnoses made by the questionnaire were compared to the diagnoses made by the daily rating calendar. The percentage of women diagnosed with PMS or PMDD was very similar for both tools (PSST and the daily symptoms calendar) in the nondepressed group (Figure 4). However, differences appeared in the group of women with depression. Both PMS (*χ*^2^(1) = 5.486; *p* < 0.05) and PMDD (*χ*^2^(1) = 35.282; *p* < 0.001) were over-diagnosed in women with depression when measured by the PSST. This means that the PSST, unlike the prospective tools, is more likely to suggest a distorted diagnosis under the influence of depression.

Depression was also found to have an impact on the quality of the diagnosis by the PSST questionnaire. Table 2 shows the specificity, sensitivity, positive predictive value (PPV), and negative predictive value (NPV) for all the analyzed groups. The PSST showed a higher sensitivity for PMS and PMDD for women with depression than for those without depression. For the specificity index, however, the results were the opposite (it was higher for the nondepressed group).

## 4. Discussion

Most PMS/PMDD studies exclude women with a diagnosis of depression. In our study, we attempted to verify how much depression changes the results of a PMS diagnosis and which PMS symptoms assessed by a questionnaire are most sensitive to mood disorders.

The present study showed that insomnia cannot be considered a symptom of PMS and PMDD. According to our results, it had a very low discriminatory power in the IRT analysis, and not many respondents reported its occurrence. For the participants experiencing insomnia, it turned out to be completely unrelated to either PMS or depression. In the study evaluating the representativeness of PMDD symptoms, insomnia was listed as a PMS symptom but never met the criterion for premenstrual changes [34]. Such an outcome contradicts the results obtained by Nicolau et al. [35]. In their study, women with PMS had a higher Insomnia Severity Index score than women without PMS, although women with PMS showed only subthreshold insomnia. Moreover, the sleep disorders were diagnosed using polysomnography, which makes it difficult to determine whether insomnia was of such a low intensity that women simply did not declare it in the self-reported inventory. However, it is worth emphasizing that the study done by Nicolau et al. [35] did not include a diagnosis of depression. Our IRT analysis showed that, among depressed women, there were more cases of mild insomnia.

Notably, the IRT analysis revealed that, for some symptoms (namely irritation, tearfulness, and overeating/food cravings), women were more confident in their answers—they either reported not having the symptoms at all or having them in a severe form and rarely chose the middle ratings. Interestingly, irritation obtained very low discrimination and extremity parameters, suggesting that it may occur in the luteal phase regardless of whether or not a woman experiences PMS/PMDD. Freeman et al. [3] hypothesized that a low discriminatory index of irritation may be due to its high correlation with other items. The IRT analysis excludes this possibility, which means that, in our study, irritability was reported by most women regardless of their PMS status.

Freeman et al. [3] found that the core symptoms of PMS include food cravings, decreased interest in activities, mood swings, cramps, aches, and anxiety/tension. Their study, however, was based on logistic regression, which only allows for assessing symptoms based on one parameter: the odds ratio. The IRT analysis uses two parameters: discrimination and difficulty. On this basis, we were able to find that fatigue/lack of energy, depressed mood, feeling overwhelmed or out of control, anxiety/tension, and decreased interest in everyday activities were the best predictors of PMS/PMDD. Food cravings had a very high discrimination parameter, although women declared its occurrence even with mild PMS intensity. In contrast to Freeman et al. [3], we found that depressed mood was a significant predictor of PMS/PMDD and had very good discriminatory and extremity parameters in the nondepressed group. A possible explanation of the differences between the studies may be that, in the case of logistic regression, the high correlation between depressed mood and mood swings meant that it did not add anything significant to the prediction accuracy of PMS. Therefore, we cannot agree that depressed mood has no primary role in pure PMS/PMDD [3]. According to our IRT analysis, depressed mood should still be considered a core symptom of PMS/PMDD.

Another goal of our study was to verify how depression affects the diagnosis of premenstrual syndrome. Specifically, we were interested in how depression affects the bias of the PSST results. The differences in item functioning (DIF) between women with and without depression were present in the cases of anger/irritation, anxiety/tension, increased sensitivity, difficulties with concentration, sleep disturbances, and physical symptoms. Moreover, women with depression reported these symptoms at lower levels of premenstrual syndrome. We also found that the retrospective method was more sensitive to depression than the daily symptoms calendar. When comparing the diagnoses made by the prospective method and PSST, it turned out that the overdiagnosis of PMS and PMDD occurred only in the group of women with depression (Figure 4). In this group, the PSST showed a higher sensitivity to PMS/PMDD with a lower specificity. This roughly indicates that, in depressive women, some symptoms may be mistakenly assessed as PMS symptoms. The daily symptoms calendar diagnosed the same proportion of women with PMS and PMDD in both the depressed and nondepressed groups. Therefore, the issues with retrospective methods include not only recall bias [4] but, also, the fact that they may be overly sensitive to mood disorders. Other studies indicate that the retrospective methods influence the overdiagnosis of PMS/PMDD [17,36]. However, the overdiagnosis in these studies was in women without depression. Therefore, it is worth paying attention to the method of depression diagnosis and the phase in which we examined the participants. In the luteal phase of the cycle, significantly more women were diagnosed with PMS by retrospective PSST. In contrast, Henz et al. [15] showed that the PSST only over-diagnosed PMDD, and the daily symptom calendar over-diagnosed PMS. The lack of information about the phase of the cycle in which the women were tested may have an impact on the obtained results.

The results of the present study may serve as practical guidelines for both clinicians and researchers. The daily calendar measured symptoms across three phases of the cycle, which allowed us to distinguish between women with PMS and those who had MD with exacerbating symptoms during the luteal phase. Retrospective methods can only be used for screening but not as a basis for the full diagnosis of PMS.

A strength of our study was the use of the prospective daily symptoms calendar for the diagnosis of PMS/PMDD, as well as the fact that we compared the results between women with and without depression. Although the findings contribute to the literature on psychological measurements of premenstrual distress, they are not without limitations. First, the current sample comprised a highly homogenous group of women at low social risk living in an urban area. Due to the study design, all participating women were required to have direct access to a computer with an internet connection. Second, we did not exclude women taking oral contraception (OC). This selection approach was taken due to the fact that we treated OC as an independent variable. Notably, the descriptive analyses revealed that taking OC did not differentiate women in terms of PMS/PMDD. Similarly, in a huge population study (*n* = 1246), the prevalence rate of PMDD did not differ between OC recipients and non-OC users [9]. Therefore, we may assume that taking OC had no effect on the analyzed phenomena in this study. Third, we did not diagnose anxiety, neurological disorders, or any other disorders that could also affect the diagnosis of PMS. Some of our participants were under the care of a psychiatrist and psychotherapist. Future studies should verify if our results are specific for depression or, rather, apply to broader emotional disorders. Another limitation was the high attrition rate. In such studies, it is hard to keep participants from dropping out, as they are asked to fill in the menstrual calendar every single day for two consecutive cycles. Women who dropped out of the study might have shared certain characteristics, such as overly low or overly high severities of the symptoms. Moreover, the level of engagement in such lengthy studies might be lower for those with mental health problems [37].

Overall, despite these limitations, the current study adds to the existing research of premenstrual disorders. To the best of our knowledge, this is the first PMS study based on an IRT analysis. The retrospective method is more sensitive to distortions resulting from both affective disorders and the phase of the cycle in which it is completed. Future research should be focused on verifying the relevance of the results obtained in other groups of women, possibly of different socioeconomic and cultural backgrounds.

## Figures and Tables

**Figure 1 life-11-01278-f001:**
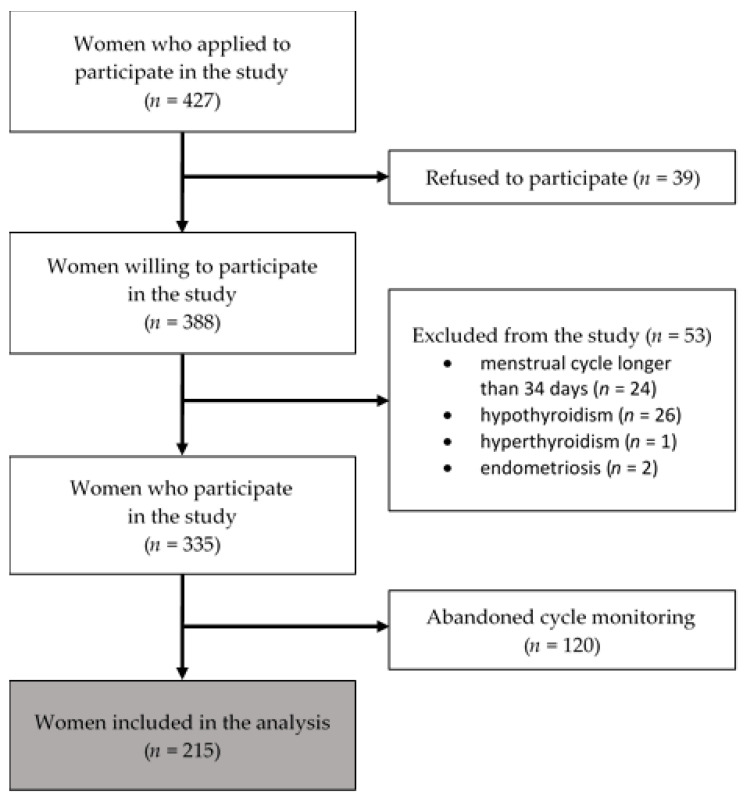
Flow diagram of the recruitment of the study population.

**Figure 2 life-11-01278-f002:**
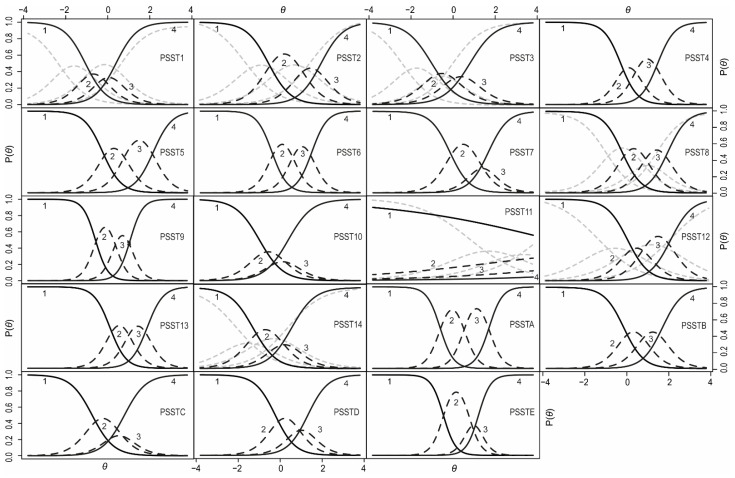
Trace curves for both parts of the PSST in the nondepressed group—symptom parts (PSST1-PSST14) and interference with daily activities (PSSTA-PSSTE). The symbol *θ* indicates PMS/PMDD on the horizontal axis. The probability level of giving each questionnaire answer is shown on the vertical axis. Four curves on the plot represent each possible response to the item: curve 1 for the “Not at all” response, curve 2 for the “Mild” response, curve 3 for the “Moderate” response, and curve 4 for the “Severe” response. A person’s PMS/PMDD level is denoted by *θ*, and it is plotted along the horizontal axis. The vertical axis shows the probability P (*θ*) of each response given for the PMS level. The curve for option 1 is high at the lowest ability level and declines gradually as the level of PMS increases. Similarly, the probability of the 4 responses is very low at low PMS/PMDD levels (*θ*) but rises as *θ* increases. The probability of the other options rises with *θ* to a certain point and then declines again. Gray dashed lines represent results from the group of women with depression. They were placed only for the items in which a significant difference (DIF) was demonstrated.

**Figure 3 life-11-01278-f003:**
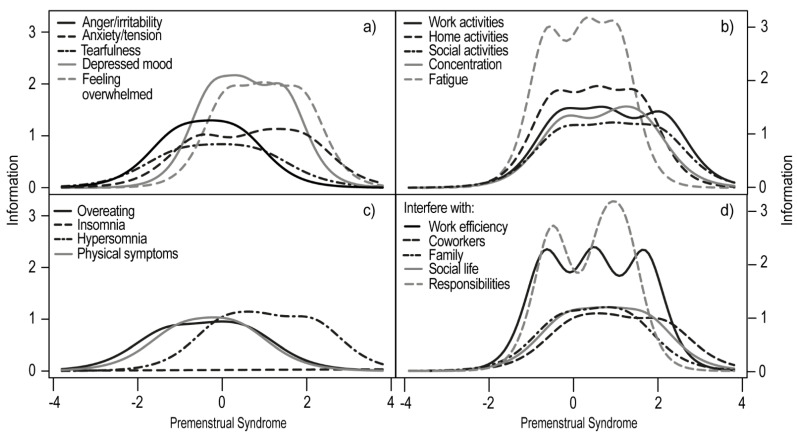
The Item Information Curves are divided into three domains for clarity (**a**–**c**) and the interference with daily activities domain (**d**). The higher the curve in the chart, the greater the discrimination factor of the item. The more it is shifted to the right, the higher difficulty parameter it represents (fewer women without PMS answered, according to the key).

**Figure 4 life-11-01278-f004:**
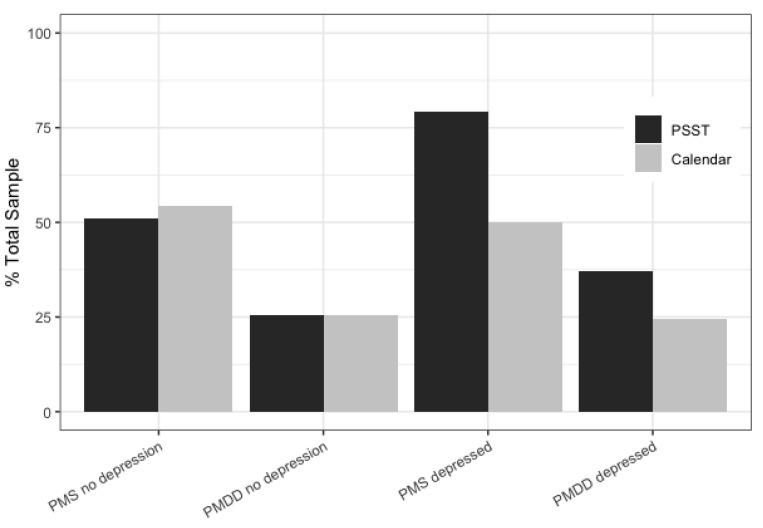
Percentage of women with PMS and PMDD diagnosed by the prospective calendar of daily symptoms or retrospective PSST among women with and without depression.

**Table 1 life-11-01278-t001:** Item discrimination and extremity parameters for women with and without depression and for the entire sample.

	Discrimination Level			Extremity Parameters								
	General	Depressed	Non-Depressed	Extrmt1			Extrmt2			Extrmt3		
				General	Depressed	Non-Depressed	General	Depressed	Non-Depressed	General	Depressed	Non-Depressed
1. Anger/irritability	1.409	1.519	2.010	−2.266	−2.296	−1.064	−0.795	−0.879	−0.290	0.636	0.599	0.391
2. Anxiety/tension	1.477	1.284	1.909	−1.373	−1.645	−0.546	0.094	−0.087	0.954	1.310	1.479	1.950
3.Tearful/sensitivity to rejection	1.681	1.429	1.655	−1.876	−2.380	−1.006	−0.806	−1.033	−0.046	0.168	−0.012	0.804
4. Depressed mood	2.042	1.898	2.521	−1.265	−2.120	−0.253	−0.286	−0.664	0.511	0.672	−0.476	1.502
5. Interest in work activities	2.097	1.849	2.271	−0.907	−1.286	−0.244	0.286	−0.163	0.828	1.256	1.114	2.171
6. Interest in home activities	2.361	2.132	2.779	−1.014	−1.300	−0.397	0.078	−0.127	0.585	0.975	0.982	1.506
7. Interest in social activities	2.009	1.563	2.174	−0.784	−0.799	−0.152	0.301	0.305	1.109	1.097	1.317	1.676
8. Concentration	1.761	1.631	2.203	−0.787	−0.941	−0.184	0.558	0.595	0.920	1.625	1.457	1.982
9. Fatigue/lack of energy	2.353	2.599	3.392	−1.580	−1.721	−0.578	−0.357	−0.388	0.348	0.735	0.889	1.084
10. Overeating/food cravings	0.988	0.998	1.784	−1.625	−1.498	−0.985	−0.302	−0.360	−0.151	0.717	1.033	−0.381
11. Insomnia	0.616	0.821	0.256	0.905	0.814	4.703	3.198	2.600	10.116	6.184	4.161	15.782
12. Hypersomnia	1.353	1.020	2.027	−0.885	−1.353	0.075	0.275	0.249	0.910	1.691	2.044	2.113
13. Feeling overwhelmed	2.233	2.368	2.618	−0.863	−0.846	0.122	0.062	−0.190	1.016	0.843	0.635	1.893
14. Physical symptoms	1.318	1.195	1.818	−2.076	−2.037	−1.284	−0.794	−0.948	−0.130	0.457	0.352	0.538
A. Interfere with work efficiency	2.253	2.528	3.175	−1.197	−1.161	−0.608	0.066	−0.177	0.531	1.132	0.735	1.710
B. Interfere with relationships with coworkers	1.930	1.794	2.136	−0.722	−0.780	−0.110	0.395	0.279	0.804	1.449	1.547	1.710
C. Interfere with family relationships	1.775	1.565	1.957	−1.644	−1.815	−0.739	−0.566	−0.995	0.279	0.480	0.272	0.796
D. Interfere with social life	2.380	2.480	2.109	−1.105	−1.316	−0.237	0.013	−0.045	0.707	0.884	0.992	1.319
E. Interfere with home responsibilities	2.424	1.930	3.492	−1.131	−1.585	−0.450	0.182	−0.125	0.767	1.063	0.980	1.215

Column Extrmt1 contains a latent trait score for which a person has a 50% chance of choosing a “Not at all” response vs. the other options. Extrmt2 shows a score that yields a 50% chance of selecting a “Not at all” or “Mild” response vs. “Moderate” or “Severe”. Extrmt3 shows the value of a latent trait for which there is a 50% chance of choosing the answer “Moderate” or lower vs. “severe”.

**Table 2 life-11-01278-t002:** Specificity, sensitivity, positive predictive value, and negative predictive value for PSST.

	PMS			PMDD		
	Whole Sample	Depressed	Non-Depressed	Whole Sample	Depressed	Non-Depressed
Specificity	40% (30, 51)	21% (10, 36)	68% (46, 85)	75% (68, 81)	71% (59, 82)	73% (58, 85)
Sensitivity	75% (66, 82)	79% (64, 90)	67% (47, 83)	51% (35, 67)	65% (41, 85)	14% (00, 58)
PPV	63% (54, 70)	50% (38, 62)	71% (51, 87)	32% (21, 45)	41% (24, 59)	0.07% (00, 34)
NPV	54% (42, 67)	50% (26, 74)	63% (42, 81)	87% (80, 92)	87% (75, 95)	85% (71, 94)

The values given in parentheses represent the 95% confidence interval values. PPV—positive predictive value; NPV—negative predictive value.

## Data Availability

The data presented in this study are openly available in the OSF repository at https://doi.org/10.17605/OSF.IO/WZDQR (accessed on 19 November 2021).

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
