# Peer review of "The Influence of Depression on Biased Diagnosis of Premenstrual Syndrome and Premenstrual Dysphoric Disorder by the PSST Inventory"

_life, 2021, doi:10.3390/life11111278_

Round 1
Reviewer 1 Report
This study aimed at evaluating the Premenstrual Symptoms Screening Tool (PSST) and the influence of major depression comorbidity with regards to the diagnosis of premenstrual syndrome (PMS) and premenstrual dysphoric disorder (PMDD). It addresses the issue of psychiatric comorbidity in diagnosing PMS and PMDD, with potential implications for both clinical practice and research in the field of premenstrual disorders. While the comparison between the PSST and a daily symptom calendar in terms of PMS and PMDD diagnoses is not novel, additional information is provided considering the discriminatory value of the PSST individual items and the influence of depressive comorbidity.
Overall, this work is of interest to the field but lacks scientific rigor. In terms of statistics, the authors must provide indicators of statistical significance and effect size in order to make reasonable conclusions about their findings. In addition, substantial effort is needed to improve the clarity of the methods section and the manuscript should be double-checked for spelling and grammatical errors.
I had previously provided a peer review of this manuscript for another MDPI journal and was surprised to have now received it in its original version (even excluding corrections of English spelling and grammar errors), while substantial efforts had been made by the authors to improve its quality during the revision process.
I would kindly ask the authors to submit the latest version of their work, or refer to their previous revisions.
Author Response
Reviewer 1: I had previously provided a peer review of this manuscript for another MDPI journal and was surprised to have now received it in its original version (even excluding corrections of English spelling and grammar errors), while substantial efforts had been made by the authors to improve its quality during the revision process.
Answer: The article was submitted to Behavioral Science. We were forced to withdraw it due to the fact that a second reviewer insisted that she/he would not accept the article until we had removed the women taking oral contraception from the sample. We provided arguments related to the fact that many studies take into account women with OC; that OC did not reduce PMS symptoms and that OC was shown to be of low effectiveness in reducing symptoms, but we did not manage to change his/her decision.
We have restored the latest version, including revisions made by a native speaker. We agree with the reviewer that all corrections have definitely improved the quality of the article. We have added all the corrections in "track changes" mode to make them more visible.
Reviewer 1: Overall, this work is of interest to the field but lacks scientific rigor. In terms of statistics, the authors must provide indicators of statistical significance and effect size in order to make reasonable conclusions about their findings.
Answer: Hopefully, the restored version meets the criteria of scientific rigor. If allowed, we would like to refer to the last comments we received from the reviewer in the previous review. After the last comment, we understood that it was neither about adding significant differences to the figures, nor a single specific statistic to this figure. We added the results of the comparison of the depressive groups in terms of the calendar diagnosis and the PSST. Now the statement is (line 458): "Both PMS (c2(1)=5.486; p<0.05) and PMDD (c2(1)=35.282; p<0.001) were overdiagnosed in women with depression when measured by PSST"
Reviewer 1: In addition, substantial effort is needed to improve the clarity of the methods section and the manuscript should be double-checked for spelling and grammatical errors.
Answer: The methods section has been divided into prospective and retrospective tools, as well as those used for the depression diagnosis. The paper has also been proofread by a native speaker.
Reviewer 1: I would kindly ask the authors to submit the latest version of their work, or refer to their previous revisions.
Answer: The version that includes corrections (adding statistics comparing PSST and calendar) has now been provided.
Reviewer 2 Report
This is an observational study on influence of depression on biased diagnosis of premenstrual syndrome and premenstrual dysphoric disorder by the PSST inventory. However, there are some serious limitations in the manuscript, which disqualify this manuscript from publication in Life.
First of all, authors should Authors should clarify what is the aim of this study. What can be the practical application of the results. What is the novelty of the conducted research ? There are many studies in the literature assessing the relationship between PMS and lifestyle factors (for example dietary habits, physical activity). Authors do not mention it at all. It would be worth at least to compare the results to the participants' BMI.
The study group is not well described. Information about contraceptive methods is missing from the inclusion criteria. Women using contraception should be definitely excluded from this study. Authors mentioned that the advantage of the study is the inclusion of women taking contraception. I don't agree and I think that this is a big mistake.
Another serious objection to this article concerns the age of the participants. Why it was 20-35? Reproductive age is to 40 or 45 years. I see no explanation for this approach
I also recommend the language correction of this manuscript.
Author Response
Reviewer 2: First of all, authors should Authors should clarify what is the aim of this study. What can be the practical application of the results. What is the novelty of the conducted research ? There are many studies in the literature assessing the relationship between PMS and lifestyle factors (for example dietary habits, physical activity). Authors do not mention it at all. It would be worth at least to compare the results to the participants' BMI.
Answer: We would like to thank the reviewer for this comment. We worked on improving the description of the aim of the study (line 102) by dividing the section into main and secondary aims. We have also included our hypotheses in that section. Referring to the novelty of our research, we clarify it in the last sentence (line 545).
However, when referring to factors influencing PMS, such as one’s lifestyle, we think that they are completely unrelated to our study. Our intention was to verify how depression distorts the diagnosis of PMS using two different tools. We could have of course taken various factors into account, but they would do little to compare the results of the two tools between the two groups, i.e., women with and without depression. This does not mean that the factors mentioned by the reviewer are irrelevant. In this study, however, they do not apply.
Reviewer 2: The study group is not well described. Information about contraceptive methods is missing from the inclusion criteria. Women using contraception should be definitely excluded from this study. Authors mentioned that the advantage of the study is the inclusion of women taking contraception. I don't agree and I think that this is a big mistake.
Answer: The inclusion of women taking OC was mentioned as the second limitation of our study (line 529). We have never stated that this is an advantage. However, we do not consider the inclusion of women with OC to be a mistake. Many studies have included women with OC in their analyses. As we wrote in the article, women with OC do not differ in the diagnosis of PMS/PMDD from women without OC. This is, in fact, a limitation of the study, as it makes our sample a non-homogeneous group, but, in our opinion, it does not automatically disqualify the entire study.
Reviewer 2: Another serious objection to this article concerns the age of the participants. Why it was 20-35? Reproductive age is to 40 or 45 years. I see no explanation for this approach
Answer: Most PMS/PMDD research is limited to this age range. At this age, PMS symptoms are the most intense.
Reviewer 2: I also recommend the language correction of this manuscript.
Answer: The article has been proofread by a native speaker.
Reviewer 3 Report
This is an original empirical study of the clinical ratings in patients with depression and co-ocurring premenstrual dysphoric disorder. The authors employed Item Response Theory to analyse the psychometric properties of different assessment tools. In my view the study design, and the performed procedures are sound enough to merit publication.
One minor suggestion for revision would be to outline more carefully a paragraph of 200-300 words in Conclusion section to highlight the core tips of the study.
Author Response
Reviewer 3: One minor suggestion for revision would be to outline more carefully a paragraph of 200-300 words in Conclusion section to highlight the core tips of the study.
Answer: We would like to thank the reviewer for this suggestion. We added a separate conclusions paragraph at the end of the article. It summarizes the results obtained in the study.
Round 2
Reviewer 1 Report
The authors have provided satisfactory answers to my concerns.
I believe the manuscript in its present form will make a valuable contribution to the field.
Reviewer 2 Report
In my last review a wrote that in this article there are to many serious limitations, which disqualify this manuscript from publication in Life. I am not agree for accept this manuscript.
This manuscript is a resubmission of an earlier submission. The following is a list of the peer review reports and author responses from that submission.
Round 1
Reviewer 1 Report
Review report
The manuscript consists in the evaluation of the Premenstrual Symptoms Screening Tool (PSST) and the influence of major depression comorbidity with regards to the diagnosis of premenstrual syndrome (PMS) and premenstrual dysphoric disorder (PMDD). It addresses the issue of psychiatric comorbidity in diagnosing PMS and PMDD, with potential implications for both clinical practice and research in the field of premenstrual disorders. While the comparison between the PSST and a daily symptom calendar in terms of PMS and PMDD diagnoses is not novel, additional information is provided considering the discriminatory value of the PSST individual items and the influence of depressive comorbidity.
Overall, this work is of interest to the field but lacks scientific rigor in terms of how to report the statistics behind the results. The authors must provide indicators of statistical significance and effect size in order to make reasonable conclusions about their findings. In addition, substantial effort is needed to improve the clarity of the methods section.
Some points detailed in the following paragraphs require attention, and are provided to help the authors revise their manuscript.
Major comments:
- Abstract: one or two sentences presenting the rationale/background of the study should be added.
- The definition of PMDD symptoms (lines 36-38) appears too restrictive. Specify that the cited symptoms are ones among others, or add a more inclusive list of symptoms (e.g. affective, cognitive and physical symptoms).
- The introduction section appears to lack a clearly defined hypothesis.
- The aims of the study are clearly defined and appear to match the analyses presented in the paper. However, whether there is a main aim and secondary aims, and if so, which one of the different listed aims is the main goal of the study, is unclear. The title indicates that the influence of depression was the main focus of the study, while the paragraph presenting the aims primarily draws attention to the discriminatory value of the individual PSST items, and secondarily to the influence of depression. This paragraph would benefit from some reformulations.
- The list of exclusion criteria (lines 100-101) does not include the presence of mental and neurological disorders (except depression and PMDD), current psychoactive treatment, or on-going pregnancy. Were women excluded if they met the criteria for any brain-related disorder? Also, the use of oral contraceptives, impacting premenstrual symptoms, is not listed as an exclusion criterion. If these aspects were not accounted for in the analyses, they should be mentioned as limitations of the study. Ideally, the authors should provide additional evidence indicating that the results are not impacted by these variables.
- The authors specify that women were randomly assigned to groups assessed either in the follicular or luteal phase using the PSST (lines 105-106 and 151). However, information is missing about how these groups were considered in the analyses. For instance, it is not clear whether these groups based on menstrual cycle phase were analysed together or not, if they were compared, or if the menstrual cycle phase was accounted for in the analyses. Also, were the whole follicular and the whole luteal phase considered? This is of relevance, as the hormonal milieu varies within these 2 phases.
- What is the relevance of providing information about the number of children the participants have (lines 111–112)? If relevant, this information would fit best in the first paragraph of the results section.
- Is there a rationale for using the SCID-I as a diagnostic tool instead of the mini-international neuropsychiatric interview (MINI), and the CES-D as a depression rating questionnaire instead of the MADRS or BDI, for example? In addition, how was the CES-D questionnaire used in this study? And who administered the questionnaires?
- Lines 122-123, the authors state “a diagnosis of depression occurring at the time of the study as well as in the past was carried out”. Please specify if the women that had depression in the past, without a current diagnosis of depression were assigned to the group of women with depression.
- Line 145, was the whole luteal phase considered for the diagnosis of PMS (i.e. at least 2 days with at least 4 moderate/severe symptoms from day 14 to day 0)?
- Line 146, please define “middle of the cycle”. Was ovulation monitored? Or does it refer to day 14 according to the last menses?
- Line 176, please specify here from which software (R?) the ltm, msm and polycor packages are from.
- From the 2.3. section of the materials and methods, it is not clear which analyses were done to answer which of the aims of the study. Please clarify.
- The result section lacks indicators of statistical significance and effect size. It is important that the authors provide information about which tests were performed, and what statistical significance was reached. For instance, the first statement on the absence of differences between women with- and without depression in terms of cycle length, age, and number of children is not supported by any numbers (lines 191-192). The authors should provide descriptive characteristics, including the results of the statistical analyses indicating group differences between women with- and without depression, in the Table S1.
- Some variables presented in table S1 are not mentioned in the text (i.e. psychiatric treatment and psychotherapy). Conversely, information about the number of women taking oral contraceptives in each group is missing in Table S1. Also, how were the PMS and PMDD groups presented in Table S1 defined (diagnosis based on daily calendars or PSST)?
- It is important that the authors provide a description of the psychoactive treatments included in their sample. This variable should be accounted for in the analysis of differences between women with- and without depression.
- The authors indicate that there was a significant difference in the extent of taking oral contraceptives, between women with- and without depression (line 192-193). Although the use of OC did not seem to affect PMS/PMDD diagnosis, this variable should be accounted for in the analysis of differences between women with- and without depression.
- From the 3.1. section of the results, it is not clear whether the discriminative value of the PSST items was based on PMS-only versus non-PMS, or PMS+PMDD versus non-PMS/PMDD. Please clarify this, and provide statistics regarding the decreased discriminatory value of PSST items in the depression group (Lines 215-216 and Table 1), and the differences in items ratings between women with- and without depression (lines 259-270).
- In Figure 2, PSST item 12 shows grey dashed lines indicating differences between women with- and without depression, according to figure caption (lines 237-238). However, PSST item 12 is not mentioned in the text reporting differences between the groups (line 257).
- In Figure 3, how were the categories of symptoms defined? The boundaries between categories do not appear to be related to any common definition of these important aspects. One would expect to find “fatigue” among physical symptoms for example.
- The section 3.2. should be completed with statistics, indicating the significance of the reported results. Significant differences between the groups and the diagnostic tools should be indicated in Figure 4.
- Lines 377-388, the discussion of the difference between the PSST and the calendar tool does not include any comparison with previous studies comparing the PSST and other prospective calendars of premenstrual symptoms (for example Henz et al., 2018). This should be added.
- The discussion is primarily focused on PMS, while the results of the present study are based on both PMS and PMDD. What is the reason for that?
- Line 390, please clarify “the way we diagnosed depression and PMS”.
- Please elaborate on the justification for not excluding women taking oral contraceptives (lines 397-398). While the authors specify that the use of oral contraceptives did not differentiate women regarding PMS and PMDD, the difference in oral contraceptive use between women with- and without depression may have influenced the results.
- As mentioned above, the limitation section should mention that some women were taking psychoactive treatment, potentially influencing the results.
Minor comments:
- Numbers are sometimes spelled out, sometimes not, without apparent consistency.
- Likewise, the authors should define each abbreviation once and then use the abbreviated forms. A few abbreviations are not defined at all (e.g. AIC and BIC, lines 205-207).
- Table S1 would add more valuable information to the main article than Figure 1. These could be interchanged.
- Line 23: “following the menstrual period” is not strictly correct, as for some women, the symptoms disappear during the menstrual period.
- Line 25: a comma is missing after “cramps”.
- Introduction: “right before menstruation” (line 38) seems too unspecific. Please specify.
- Line 165, “on the contrary” is used, while there is no apparent opposition with the previous statement (both sentences about Item Response Theory).
- Line 167, please change “Item’s informative value” to “Items’ informative values” or “An item’s informative value”.
- Lines 223-228 should go into the Figure 2 caption. The authors should also check the results section for repetitions.
- I suggest reformulating “This means that” in line 244-245, by merging this sentence with the one before, for example by using “indicating” as connector.
- Line 317-318 “To calculate the sensitivity and specificity of PSST, the PMS and PMDD diagnoses made by the questionnaire were compared to the diagnoses made by the daily rating calendar.” could be used as introductive sentence for the 3.2. sub-section.
- Typo line 378, “PMSS”.
- Typo line 383, the authors write “Therefore, issues with prospective methods include not only recall bias, but also the fact that they may be overly sensitive to mood disorders.”. Here it should be “retrospective methods”.
- Line 399: “Notably, the descriptive analyses revealed that taking OC did not differentiate women in terms of PMS”. According to the first paragraph of the results section, “PMDD” should also be included in this statement.
- While the language level is fine overall, the manuscript could benefit from being checked by an English-native speaker.
Author Response
We would like to thank you for all the comments on our article. Based on your recommendations we have incorporated several changes, and we believe that these changes have improved the quality of the revised manuscript:
Major comments:
- Abstract: one or two sentences presenting the rationale/background of the study should be added.
We have added the background sentence as the first sentence in the abstract.
- The definition of PMDD symptoms (lines 36-38) appears too restrictive. Specify that the cited symptoms are ones among others, or add a more inclusive list of symptoms (e.g. affective, cognitive and physical symptoms).
We thank the reviewer for this suggestion. We have changed this paragraph as follows: “PMDD is characterized by marked affective, cognitive and physical symptoms that occur repeatedly before menstruation and remit at the onset of, or shortly after menses”
- The introduction section appears to lack a clearly defined hypothesis.
We thank the reviewer for bringing this to our attention. We have added the following hypothesis to the introduction section: " We hypothesized that depression would affect the overdiagnosis of PMS/PMDD in both prospective and retrospective methods. (...) " We hypothesized that PSST items would have different diagnostic power and that depression would affect responses in only some of them."
- The aims of the study are clearly defined and appear to match the analyses presented in the paper. However, whether there is a main aim and secondary aims, and if so, which one of the different listed aims is the main goal of the study, is unclear. The title indicates that the influence of depression was the main focus of the study, while the paragraph presenting the aims primarily draws attention to the discriminatory value of the individual PSST items, and secondarily to the influence of depression. This paragraph would benefit from some reformulations.
We agree with this criticism. Indeed, the main goal of our research was not stated explicitly enough.
We have reformulated this paragraph by pointing out the main and secondary aims of our study.
- The list of exclusion criteria (lines 100-101) does not include the presence of mental and neurological disorders (except depression and PMDD), current psychoactive treatment, or on-going pregnancy. Were women excluded if they met the criteria for any brain-related disorder? Also, the use of oral contraceptives, impacting premenstrual symptoms, is not listed as an exclusion criterion. If these aspects were not accounted for in the analyses, they should be mentioned as limitations of the study. Ideally, the authors should provide additional evidence indicating that the results are not impacted by these variables.
We thank the reviewer for pointing this out. We have added on-going pregnancy to the exclusion criteria. However, we did not exclude women with neurological disorders or mental disorders other than MD. We have added this information to the limitations section.
- The authors specify that women were randomly assigned to groups assessed either in the follicular or luteal phase using the PSST (lines 105-106 and 151). However, information is missing about how these groups were considered in the analyses. For instance, it is not clear whether these groups based on menstrual cycle phase were analysed together or not, if they were compared, or if the menstrual cycle phase was accounted for in the analyses. Also, were the whole follicular and the whole luteal phase considered? This is of relevance, as the hormonal milieu varies within these 2 phases.
We really appreciate this suggestion. We agree with the reviewer that we have not provided enough information on this in the article. We have now expanded the description of the research procedure and added a description of analyzes taking into account the phase of the cycle.
- What is the relevance of providing information about the number of children the participants have (lines 111–112)? If relevant, this information would fit best in the first paragraph of the results section.
We thank the reviewer for pointing this out. We have added this information to better describe our study population.
- Is there a rationale for using the SCID-I as a diagnostic tool instead of the mini-international neuropsychiatric interview (MINI), and the CES-D as a depression rating questionnaire instead of the MADRS or BDI, for example? In addition, how was the CES-D questionnaire used in this study? And who administered the questionnaires?
We have used SCID-I and CES-D due to good psychometric properties of their Polish adaptations, as well as the fact that both are well known and commonly used. This enables possible replication of this research. We have used SCID-I because there is no Polish adaptation of MINI. Although a Polish adaptation of BDI-II has been recently introduced, it was still in the making at the time of designing and conducting this study. CES-D was used as a screening tool. We have attempted to keep the size of both groups (depressed and nondepressed) even. However, the final diagnosis was based on the clinical interview. We have added this information to the description of CES-D and the Procedure and participants section.
- Lines 122-123, the authors state “a diagnosis of depression occurring at the time of the study as well as in the past was carried out”. Please specify if the women that had depression in the past, without a current diagnosis of depression were assigned to the group of women with depression.
We thank the reviewer for pointing this out. We have clarified the issue as follows: “Women diagnosed with depression and those who previously experienced depression (within the last 3 years) were included in the depressed group.”
- Line 145, was the whole luteal phase considered for the diagnosis of PMS (i.e. at least 2 days with at least 4 moderate/severe symptoms from day 14 to day 0)?
Our article now reads: “Calendar scores were summed across the daily symptom reports for three five-day periods: last days of the luteal phase, first days of the menstrual cycle and the follicular phase (from 12-16 days after last menses)”. We have now specified what the middle of the cycle/follicular phase means by clarifying that it refers to scores given during 12-16 days after last menses. We hope that the five-day periods are understandable in reference to both luteal and first days of the cycle.
- Line 146, please define “middle of the cycle”. Was ovulation monitored? Or does it refer to day 14 according to the last menses?
Due to the fact that we have used scores from 5 days before menses (as a luteal phase) we used the same period of time (5 days) in the middle of the cycle. We agree that the manuscript lacked sufficient information on the matter. We have now stated in the paragraph that the middle of the cycle are the days from 12 to 16 days after last menses.
- Line 176, please specify here from which software (R?) the ltm, msm and polycor packages are from.
We thank the reviewer for pointing this out. These packages are from the R software. We have added this information to the relevant paragraph.
- From the 2.3. section of the materials and methods, it is not clear which analyses were done to answer which of the aims of the study. Please clarify.
We agree with this criticism. We have now added this information to the Statistical Analysis section: “To verify the hypothesis that depression affects the diagnosis of PMS/PMDD made by prospective and retrospective methods, we have utilized both Item Response Theory (IRT) and sensitivity/specificity calculation.”
- The result section lacks indicators of statistical significance and effect size. It is important that the authors provide information about which tests were performed, and what statistical significance was reached. For instance, the first statement on the absence of differences between women with- and without depression in terms of cycle length, age, and number of children is not supported by any numbers (lines 191-192). The authors should provide descriptive characteristics, including the results of the statistical analyses indicating group differences between women with- and without depression, in the Table S1.
We agree that the previous way of reporting the results might have been insufficient. We have added all necessary statistics to the Results section.
- Some variables presented in table S1 are not mentioned in the text (i.e. psychiatric treatment and psychotherapy). Conversely, information about the number of women taking oral contraceptives in each group is missing in Table S1. Also, how were the PMS and PMDD groups presented in Table S1 defined (diagnosis based on daily calendars or PSST)?
We thank the reviewer for bringing this to our attention. We have now updated the whole table by adding the statistics for all groups. We have also clarified how PMS and PMDD were diagnosed (by prospective methods). We also added the CES-D scores and information about taking OC.
- It is important that the authors provide a description of the psychoactive treatments included in their sample. This variable should be accounted for in the analysis of differences between women with- and without depression.
These are very important points. We asked the participants (using self-report questions) if they participated in psychotherapy and whether or not they used psychiatric care. We also asked them about their diagnoses of depression (self-report) or anxiety disorder. All participants from the nondepressed group participated in psychotherapy/psychiatric treatment due to anxiety disorders. We have now added this information to the participants' descriptions.
- The authors indicate that there was a significant difference in the extent of taking oral contraceptives, between women with- and without depression (line 192-193). Although the use of OC did not seem to affect PMS/PMDD diagnosis, this variable should be accounted for in the analysis of differences between women with- and without depression.
We thank the reviewer for this suggestion. Due to an error in the key created in the Excel file both retrospective and prospective tools diagnosed PMS/PMDD when participants met 4 criteria. Both DSM-5 and PSST instructions require at least 5 symptoms for PMS/PMDD diagnosis. Fortunately, our article stated that both diagnoses were made when only 4 criteria were met, meaning the error was noticed in time. So we recalculated the results wherever the PMS/PMDD variable was present. Because of this, the descriptive statistics have changed. However, when referring to the analysis taking OC into account, we attempted to include it in as many analyses as possible. Unfortunately, adding this variable to all analyses is impossible. Adding it causes the groups to be too small to make appropriate comparisons. Due to this fact, we have expanded the description of including OC participants in the limitations section.
- From the 3.1. section of the results, it is not clear whether the discriminative value of the PSST items was based on PMS-only versus non-PMS, or PMS+PMDD versus non-PMS/PMDD. Please clarify this, and provide statistics regarding the decreased discriminatory value of PSST items in the depression group (Lines 215-216 and Table 1), and the differences in items ratings between women with- and without depression (lines 259-270).
We thank the reviewer for these suggestions. The reason why we referred to both PMS/PMDD throughout the whole paragraph was due to the fact that we compared PMS+PMDD versus non-PMS/PMDD. This is because of how PMS and PMDD were diagnosed. If someone met the criteria for PMDD, they certainly also met the criteria for PMS.
The reviewer asked us to provide statistics regarding the decreased discriminatory value of PSST items in the depression group. However, the IRT analysis does not allow the comparison of the two groups in this respect. Therefore, the IRT analysis refers to "visual inspection" as there are also no mathematical tools for comparisons between the two items. That is why we explained it numerous times in the text (e.g.,. “On average, depressed women had higher mean scores of anger/irritation, anxiety/tension, as well as increased sensitivity. The curves for these items were skewed sharply to the left, which indicates that for lower levels of PMS, women reported the occurrence of these symptoms.”
IRT has been known in psychometry for many years and is often regarded as superior to the classical test theory. However, it was only the rapid development of computers and statistical tools that allowed for the widespread use of these analyzes. We would like to emphasize this because, in our understanding, the lack of a specific test to determine the statistical difference does not make the analysis worthless.
- In Figure 2, PSST item 12 shows grey dashed lines indicating differences between women with- and without depression, according to figure caption (lines 237-238). However, PSST item 12 is not mentioned in the text reporting differences between the groups (line 257).
The responses to this item also differed depending on whether the woman was depressed or not. We have now added it to the DIF summary (“DIF: 1, 2, 3, 8, 11, 12, and 14.”). Item 12, however, was described in the text: “Differences were also present for the questions concerning sleep disorders”.
- In Figure 3, how were the categories of symptoms defined? The boundaries between categories do not appear to be related to any common definition of these important aspects. One would expect to find “fatigue” among physical symptoms for example.
We thank the reviewer for this criticism. The division into categories resulted from the attempt to increase the transparency of the chart. 14 lines on one chart would most likely be unreadable. We wanted to keep a fairly equal number of items on each chart. We have now changed the description of the figure explaining the division of the symptoms into three separate graphs.
- The section 3.2. should be completed with statistics, indicating the significance of the reported results. Significant differences between the groups and the diagnostic tools should be indicated in Figure 4.
We appreciate this suggestion, although we feel like we cannot add any additional statistics. The differences described in this section are measured using two different nominal variables (PMS retrospective [PMSr] and PMS prospective [PMSp]). In Figure 4 there is only one answer (1 - PMS/PMDD) from both variables (we did not put the percentage of women without PMS for neither of the variables - score 0). For the comparison between PMSr and PMSp using a cross table we would need 4 groups (PMSr1PMSp0 + PMSr1PMSp1 + PMSr0PMSp0 + PMSr0PMSp1). Differences between those groups are not the difference between PMSr1 and PMSp1. When we tried to merge those variables into one, the biggest problem was that one participant may have given 2 answers at the same time (PMS in both PMSr and PMSp). We realize that the percentages do not necessarily indicate statistical differences, but we do not have a statistical model that can compare them (unless there is a method that we do not know of).
- Lines 377-388, the discussion of the difference between the PSST and the calendar tool does not include any comparison with previous studies comparing the PSST and other prospective calendars of premenstrual symptoms (for example Henz et al., 2018). This should be added.
We appreciate this suggestion. We made sure to expand this section of the manuscript.
- The discussion is primarily focused on PMS, while the results of the present study are based on both PMS and PMDD. What is the reason for that?
We thank the reviewer for pointing this out. We have reformulated the whole paragraph so that it refers not only to women with PMS but also women with PMDD.
- Line 390, please clarify “the way we diagnosed depression and PMS”.
We have clarified this as follows: “A strength of our study is the use of the prospective daily symptoms calendar for the diagnosis of PMS/PMDD”
- Please elaborate on the justification for not excluding women taking oral contraceptives (lines 397-398). While the authors specify that the use of oral contraceptives did not differentiate women regarding PMS and PMDD, the difference in oral contraceptive use between women with- and without depression may have influenced the results.
We thank the reviewer for bringing this to our attention. We agree that a better rationale for this decision should have been added. We have thus expanded the limitation section of the manuscript.
- As mentioned above, the limitation section should mention that some women were taking psychoactive treatment, potentially influencing the results.
We appreciate this suggestion. We have added this information to the limitation section.
Minor comments:
- Numbers are sometimes spelled out, sometimes not, without apparent consistency.
We have corrected this so that numbers are no longer spelled out.
- Likewise, the authors should define each abbreviation once and then use the abbreviated forms. A few abbreviations are not defined at all (e.g. AIC and BIC, lines 205-207).
We have added explanations for all abbreviations throughout the manuscript.
- Table S1 would add more valuable information to the main article than Figure 1. These could be interchanged.
We thank the reviewer for this suggestion. Since tables and charts of theIRT analysis take up a lot of space in the article, we did not want to increase the number of tables. However, the Flowchart shows the recruitment process and the high attrition rate that occurs in studies that require monitoring of two consecutive cycles. We would prefer to keep it in the manuscript as it is one of the standards of the STROBE protocol (Strobe Initiative, 2008)
- Line 23: “following the menstrual period” is not strictly correct, as for some women, the symptoms disappear during the menstrual period.
We agree with this point. We corrected it as follows: “symptoms that occur in the luteal phase of the menstrual cycle and subside at the beginning of or during the menstrual period”
- Line 25: a comma is missing after “cramps”.
A comma was added to this sentence.
- Introduction: “right before menstruation” (line 38) seems too unspecific. Please specify.
We agree with this point. We have changed the sentence to " symptoms that occur repeatedly before menstruation and remit at the onset of, or shortly after menses".
- Line 165, “on the contrary” is used, while there is no apparent opposition with the previous statement (both sentences about Item Response Theory).
We thank the reviewer for pointing this out. We have reformulated the sentence.
- Line 167, please change “Item’s informative value” to “Items’ informative values” or “An item’s informative value”.
Corrected, thank you.
- Lines 223-228 should go into the Figure 2 caption. The authors should also check the results section for repetitions.
We thank the reviewer for these suggestions. We have moved this section to the Figure 2 caption. We have also deleted all repetitions to make this paragraph more clear.
- I suggest reformulating “This means that” in line 244-245, by merging this sentence with the one before, for example by using “indicating” as connector.
We have merged the sentences according to the reviewer’s suggestions.
- Line 317-318 “To calculate the sensitivity and specificity of PSST, the PMS and PMDD diagnoses made by the questionnaire were compared to the diagnoses made by the daily rating calendar.” could be used as introductive sentence for the 3.2. sub-section.
We have moved this sentence according to the reviewer's suggestion.
- Typo line 378, “PMSS”.
We have corrected this typo.
- Typo line 383, the authors write “Therefore, issues with prospective methods include not only recall bias, but also the fact that they may be overly sensitive to mood disorders.”. Here it should be “retrospective methods”.
We thank the reviewer for pointing this out. Of course, this sentence was about retrospective methods. We corrected this sentence.
- Line 399: “Notably, the descriptive analyses revealed that taking OC did not differentiate women in terms of PMS”. According to the first paragraph of the results section, “PMDD” should also be included in this statement.
We have corrected it according to the comment no. 23 in the major issues section of the review.
- While the language level is fine overall, the manuscript could benefit from being checked by an English-native speaker.
The corrected version of the article has been proofread by a native speaker, which has hopefully improved the overall quality of the manuscript.
Reviewer 2 Report
The authors present a well-written and well-structured manuscript in which they examine the predictive utility of a retrospective screening tool for PMS/PMDD in women with and without depression. In order to do so, they additionally asked their sample to rate their daily symptoms across two consecutive menstrual cycles allowing them to prospectively diagnose their sample with PMS/PMDD (i.e., the gold standard). The complex statistical analyses used in this manuscript (Item Response Theory) are not only highly suitable for the research questions and well explained but also are first used in research on PMS/PMDD diagnosis (to the best of my knowledge). I consider this manuscript to be a valuable addition to the field.
I mostly have minor comments which I hope will improve the manuscript further:
Abstract:
- I recommend editing the abstract to more clearly delineate the impressive design and findings of the study: Define the abbreviations PMS and PMDD at first use and, for example, consistently use the terms „retrospective diagnosis with PSST“ and „prospective diagnosis with daily symptom calender“ to increase clarity. I recommend considering to include the following sentences from the result section as they summarize one of the main findings very clearly: “Both PMS and PMDD were overdiagnosed in women with depression when measured by PSST. This means that PSST, unlike prospective tools, is more likely to suggest a distorted diagnosis under the influence of depression.” (lines 307-310).
Introduction:
- Line 38: „[…] that occur repeatedly right before menstruation […]” I recommend changing the wording to “that occur repeatedly before menstruation”, given that women with PMDD can experience symptom emergence any time after ovulation (see DOI:10.1017/S0033291719000849)
Materials and Methods:
- Line 97/98: “Women aged 20-35 were invited to participate in a study for the women with or without premenstrual problems.” Please describe in more detail how the study was advertised and whether women were paid for participation.
- Line 108/109: “120 women did not complete monitoring […]”. If possible (i.e., if such information is available), please clarify the reasons the participants gave for withdrawing from study participation.
- Line 112: “[…] of them had one children […]” Please to correct to “had one child”.
- Line 119: I recommend using subtitles for the sections 2 Instruments to ease readability (e.g., Depression, Prospective PMS/PMDD Diagnosis, Retrospective PMS/PMDD Diagnosis)
- Line 121/122: “Interviewers were blind to the participants’ scores.” Please clarify which which scores are meant? The CES-D self-report scores later described in the text?
- Line 135: Please specify the 4-point Likert scale, as you later use its rating options “moderate” and “severe”.
- Line 144: Please specify the exact cycle days that constitute the five-day period of the follicular phase.
- Lines 144-149/prospective PMS/PMDD diagnosis: “For PMS diagnosis, participants had to mark at least 4 symptoms as moderate or severe for at least 2 days in the luteal phase without showing any symptoms in the middle of the cycle. Additionally, they had to point out that their symptoms interfered with their functioning at least in a moderate way. The same criteria were used for PMDD diagnosis, except that the symptoms had to be specified by participants as severe.” and Lines 157-161/retrospective PMS/PMDD diagnosis: “At least four moderate or severe symptoms from the first part with at least one being from the first four symptoms. Moreover, in the PMS group it was required that that the symptoms interfered with one’s functioning at least moderately. The same criteria were used for PMDD diagnosis, except that the symptoms and list from second part had to be specified by participants as severe”.
I am curious as to why and how the authors decided on the numerical requirement of four symptoms for PMS and PMDD diagnosis. As they correctly point out in the introduction, “there still is no unified and standardized list of criteria for PMS” (line 30/31). However, such list exists for PMDD (i.e., the DSM-5 diagnostic criteria), the DSM-5 requires five symptoms for diagnosis. Also, PMS is often diagnosed with only one symptom present. Next to the numerical cut-off of five symptoms, DSM-5 requires distress or impairment for the diagnosis. In this manuscript, the authors require both. In sum, the PMS and PMDD diagnostic criteria the authors employ are too strict regarding the number of symptoms and the presence of impairment and analyses should be repeated with adjusted criteria. - Line 158: Please delete one of the two “that”.
Results:
- Figure 4: I think it would be useful to the reader to point out (in the figure caption or legend) that PSST is the retrospective and Calendar the prospective tool with which PMS/PMDD were diagnosed
Discussion:
- Lines 348-353: “Interestingly, irritation obtained very low discrimination and extremity parameters, suggesting that it may occur in the luteal phase regardless of whether a woman experiences PMS/PMDD or not. […] which means that in our study, irritability was reported by most women regardless of their PMS status”. This finding does not align with large epidemiologic daily ratings studies indicating that most females do not show cyclical symptoms (Gehlert et al., 2009). Please provide an embedding of this result in the current literature.
- Limitations: Do the authors have any other information available regarding mental disorders (e.g., GAD), general distress, or trait-level negative affect of their sample? This question goes back to the possibility that the findings could be not specific to depression but to broader/transdiagnostic criteria.
- Line 396/397: I appreciate the discussion of not excluding women who take OCs as a limitation. However, I recommend adding a sentence describing how many women of the sample are taking OCs to the brief description of the sample in lines 111-112.
- Line 402: “Another limitation is the high attrition rate.” I am wondering, do you have any hypotheses regarding who withdrew from study participation? In other words, do you suspect systematic withdrawal from a specific group of participants (e.g., the ones who suffer more or who suffer less)? And if so, adding a brief discussion of it might be valuable.
Author Response
We would like to thank you for all the comments on our article. Based on your recommendations we have incorporated several changes, and we believe that these changes have improved the quality of the revised manuscript:
Abstract:
- I recommend editing the abstract to more clearly delineate the impressive design and findings of the study: Define the abbreviations PMS and PMDD at first use and, for example, consistently use the terms „retrospective diagnosis with PSST“ and „prospective diagnosis with daily symptom calender“ to increase clarity. I recommend considering to include the following sentences from the result section as they summarize one of the main findings very clearly: “Both PMS and PMDD were overdiagnosed in women with depression when measured by PSST. This means that PSST, unlike prospective tools, is more likely to suggest a distorted diagnosis under the influence of depression.” (lines 307-310).
We thank the reviewer for this criticism. We have corrected the abstract according to their suggestions.
Introduction:
- Line 38: „[…] that occur repeatedly right before menstruation […]” I recommend changing the wording to “that occur repeatedly before menstruation”, given that women with PMDD can experience symptom emergence any time after ovulation (see DOI:10.1017/S0033291719000849)
We agree with this suggestion. We have changed this sentence to "symptoms that occur repeatedly before menstruation and remit at the onset of, or shortly after menses".
Materials and Methods:
- Line 97/98: “Women aged 20-35 were invited to participate in a study for the women with or without premenstrual problems.” Please describe in more detail how the study was advertised and whether women were paid for participation.
We have provided detailed information about the study invitation and remuneration.
- Line 108/109: “120 women did not complete monitoring […]”. If possible (i.e., if such information is available), please clarify the reasons the participants gave for withdrawing from study participation.
We have added this information to the paragraph: “120 women did not complete monitoring of the menstrual cycle, and 53 women were excluded (2 became pregnant, and the rest withdrew their participation without giving any reasons).”
- Line 112: “[…] of them had one children […]” Please to correct to “had one child”.
Corrected, thank you.
- Line 119: I recommend using subtitles for the sections 2 Instruments to ease readability (e.g., Depression, Prospective PMS/PMDD Diagnosis, Retrospective PMS/PMDD Diagnosis)
We thank the reviewer for this suggestion. We have added subtitles to this section which has significantly improved its clarity.
- Line 121/122: “Interviewers were blind to the participants’ scores.” Please clarify which scores are meant? The CES-D self-report scores later described in the text?
We thank the reviewer for pointing this out. We have clarified as follows: “Interviewers were blind to all of the participants’ scores”. We have also improved on the description of the procedure, making it easier to follow. Moreover, we have described how CES-D was used (in the procedure paragraph and the description of CES-D). Lastly, we have added the CES-D scores to the descriptive statistics table.
- Line 135: Please specify the 4-point Likert scale, as you later use its rating options “moderate” and “severe”.
We thank the reviewer for bringing it to our attention. In fact, it was not the Likert scale. Therefore, we have changed this sentence to: “Participants rated the daily severity of symptoms on a 4-point scale.”
- Line 144: Please specify the exact cycle days that constitute the five-day period of the follicular phase.
We have added that the middle of the cycle in our study are the days from 12 to 16 after last menses.
- Lines 144-149/prospective PMS/PMDD diagnosis: “For PMS diagnosis, participants had to mark at least 4 symptoms as moderate or severe for at least 2 days in the luteal phase without showing any symptoms in the middle of the cycle. Additionally, they had to point out that their symptoms interfered with their functioning at least in a moderate way. The same criteria were used for PMDD diagnosis, except that the symptoms had to be specified by participants as severe.” and Lines 157-161/retrospective PMS/PMDD diagnosis: “At least four moderate or severe symptoms from the first part with at least one being from the first four symptoms. Moreover, in the PMS group it was required that that the symptoms interfered with one’s functioning at least moderately. The same criteria were used for PMDD diagnosis, except that the symptoms and list from second part had to be specified by participants as severe”.
I am curious as to why and how the authors decided on the numerical requirement of four symptoms for PMS and PMDD diagnosis. As they correctly point out in the introduction, “there still is no unified and standardized list of criteria for PMS” (line 30/31). However, such list exists for PMDD (i.e., the DSM-5 diagnostic criteria), the DSM-5 requires five symptoms for diagnosis. Also, PMS is often diagnosed with only one symptom present. Next to the numerical cut-off of five symptoms, DSM-5 requires distress or impairment for the diagnosis. In this manuscript, the authors require both. In sum, the PMS and PMDD diagnostic criteria the authors employ are too strict regarding the number of symptoms and the presence of impairment and analyses should be repeated with adjusted criteria.
We really appreciate the reviewer for pointing this out. Unfortunately, this choice is not at all justified. It results from an error that occurred while creating the formula for converting the results in the Excel sheet. Fortunately, the description of the instruments (PSST and calendar) was made on the basis of the formula, not the DSM-5 criteria and the PSST instruction. Thank you very much for catching this error, because diagnoses based on incorrect criteria would render this study useless. We have changed the Excel formula and recalcuted all statistics in which both PMS and PMDD variables were used. We also changed the table with descriptive statistics. The new PMS/PMDD diagnosis did not affect any IRT analyzes due to the fact that IRT analyzes utilized the probability of a latent trait rather than the diagnosis of PMS.
- Line 158: Please delete one of the two “that”.
Corrected, thank you. The manuscript has also been read and corrected by a native speaker.
Results:
- Figure 4: I think it would be useful to the reader to point out (in the figure caption or legend) that PSST is the retrospective and Calendar the prospective tool with which PMS/PMDD were diagnosed
We agree with the reviewer’s suggestion. We have added this information to the caption of Figure 4.
Discussion:
- Lines 348-353: “Interestingly, irritation obtained very low discrimination and extremity parameters, suggesting that it may occur in the luteal phase regardless of whether a woman experiences PMS/PMDD or not. […] which means that in our study, irritability was reported by most women regardless of their PMS status”. This finding does not align with large epidemiologic daily ratings studies indicating that most females do not show cyclical symptoms (Gehlert et al., 2009). Please provide an embedding of this result in the current literature.
We thank the reviewer for this suggestion. We are familiar with the Gehlert et al. 2009 study. It is one of the few studies that used prospective methods of diagnosis on such a large group of respondents. However, it describes only the prevalence of PMDD in the general population. The authors of this study did not break down the individual symptoms. Item Response Theory (IRT) analysis allows us to examine the style and manner of responding to each item separately. Therefore, we can analyze how women respond to each item. Women admit being irritated in the luteal phase whether or not they meet the PMS criteria.
- Limitations: Do the authors have any other information available regarding mental disorders (e.g., GAD), general distress, or trait-level negative affect of their sample? This question goes back to the possibility that the findings could be not specific to depression but to broader/transdiagnostic criteria.
We thank the reviewer for pointing this out. We have expanded our limitation section regarding this topic.
- Line 396/397: I appreciate the discussion of not excluding women who take OCs as a limitation. However, I recommend adding a sentence describing how many women of the sample are taking OCs to the brief description of the sample in lines 111-112.
We appreciate this suggestion. We have added this information to both the sample description and Table S1 with descriptive statistics.
- Line 402: “Another limitation is the high attrition rate.” I am wondering, do you have any hypotheses regarding who withdrew from study participation? In other words, do you suspect systematic withdrawal from a specific group of participants (e.g., the ones who suffer more or who suffer less)? And if so, adding a brief discussion of it might be valuable.
We thank the reviewer for this suggestion. We have added a few sentences discussing the issue in the relevant paragraph.
Round 2
Reviewer 1 Report
A good effort was made to improve the manuscript, by implementing a more rigorous approach when reporting the procedures and results. While most of my comments have been addressed, a few remain unanswered or did not receive a satisfactory explanation. Importantly, the influence of psychoactive treatments should be considered as a limitation, as well as the inclusion of women without on-going depression in the depressed group.
Major comments:
Line 126-127: “Some of the study participants used psychological or psychiatric help for a depressive disorder or anxiety. Women from the nondepressed group who participated in psychiatric care/psychotherapy have had anxiety problems.” Please specify (1) what psychiatric help means (if psychoactive treatments, please provide further information about them), (2) if participants from the non-depressed group had anxiety problems in the past, or at the time of the study, and (3) if they were medicated.
Lines 435-437: “Third, we did not diagnose anxiety, neurological disorders, or any other disorders that could also affect the diagnosis of PMS. Some of our participants were under the care of a psychiatrist and psychotherapist. Future studies should verify if our results are specific for depression or rather to broader emotional disorders.” PMDD diagnosis should also be mentioned, along with PMS. In addition, psychoactive treatments are not mentioned, despite the authors declared having added this limitation. I suspect that this was hidden under the term “care of a psychiatrist”, which can be misleading. This limitation should be stated clearly.
Line 144-146: “Women diagnosed with depression and those who previously experienced depression (within the last 3 years) were included in the depressed group.”. The fact that women without a current diagnosis of depression were included in the depressed group is a limitation of the study. This should be mentioned in the discussion. The authors could provide the proportion of women with a present diagnosis of depression within the depressed group in the limitation section to let the reader better appreciate the potential impact of this.
The authors did not provide an answer about the timing in which the PSST was filled in. Were the whole follicular and the whole luteal phase considered? Or was there a more specific time window within the follicular and the luteal phase when participants responded to the questionnaire?
The authors claim that it is not possible to run statistical analysis on the discriminative value of the PSST items. It is however still unclear to me why the comparison of numerical values indicating discrimination levels between participants with depression and non-depressed participants, as presented in Table 1, cannot be performed.
Importantly, in the case where results are only based on visual inspection, this should be stated clearly, and the authors should avoid mentioning numerical indicators, in the description of these results. For example, while it is clear that trace curve analysis (Figure 2) is based on visual inspection, the authors write “depressed women had higher mean scores of anger/irritation, anxiety/tension, and increased sensitivity”, suggesting that the results are based on numerical (as opposed to graphical) data.
Similarly, I do not understand why statistics could not assess the differences in diagnoses proportions based on the different screening tools, across the different groups as presented in Figure 4 “Percentage of women with PMS and PMDD diagnosed by the prospective calendar of daily symptoms or retrospective PSST among women with and without depression”. Chi square tests can be used for each comparison (PSST versus Calendar in each of the 4 groups).
Minor comment:
Line 91: ”We hypothesized that depression would affect the overdiagnosis of PMS/PMDD in both prospective and retrospective methods.” Please replace “affect” by “contribute to” or “lead to”.
"A description of analyzes taking into account the phase of the cycle" (from the authors replies) was not found in the new version of the manuscript.
Author Response
We would like to thank the reviewer for appreciating the changes we have made. We agree with the reviewer that the quality of the article has significantly improved. Understanding that some elements of our text are not yet fully clear, we have introduced appropriate changes so that they would not raise any additional doubts among readers.
**Lines 126-127: "Please specify (1) what psychiatric help means (if psychoactive treatments, please provide further information about them), (2) if participants from the non-depressed group had anxiety problems in the past, or at the time of the study, and (3) if they were medicated."
We have corrected the text to dispel any doubts. In line 112, we elaborated on the self-report questions. In line 126 we added the information that this is a declaration of women about using psychiatric help. Regarding the questions of the reviewer: (1) we only asked "Are you getting/have you gotten psychiatric help (in the past)?". We do not have any additional information about what kind of help it was. (2) these are the current anxiety issues (we have added this information in line 128); (3) we do not have this information.
**Lines 435-437: "PMDD diagnosis should also be mentioned, along with PMS. In addition, psychoactive treatments are not mentioned, despite the authors declared having added this limitation. I suspect that this was hidden under the term “care of a psychiatrist”, which can be misleading. This limitation should be stated clearly."
We thank the reviewer for bringing this to our attention. We have added PMDD to this statement. We have also added the information that we did not verify whether participants were taking any psychiatric medications.
**Lines 144-146: "The fact that women without a current diagnosis of depression were included in the depressed group is a limitation of the study. This should be mentioned in the discussion. The authors could provide the proportion of women with a present diagnosis of depression within the depressed group in the limitation section to let the reader better appreciate the potential impact of this."
We agree with the reviewer that this is very important information. We have now added it in line 148. We have also verified whether women with past and current depression differed in terms of their CES-D scores.
**The authors did not provide an answer about the timing in which the PSST was filled in. Were the whole follicular and the whole luteal phase considered? Or was there a more specific time window within the follicular and the luteal phase when participants responded to the questionnaire?
The follicular and luteal phases were defined in the same way for prospective and retrospective methods. For better clarity, we have now added this information to the paragraph that describes PSST.
**The authors claim that it is not possible to run statistical analysis on the discriminative value of the PSST items. It is however still unclear to me why the comparison of numerical values indicating discrimination levels between participants with depression and non-depressed participants, as presented in Table 1, cannot be performed.
IRT analysis allows us to see how the participants respond to each questionnaire item. It also allows us to check their discrimination parameters and (with the help of the DIF analysis) whether variables such as depression change the way of answering. Just as the factor loadings in EFA cannot be statistically compared, the IRT parameters cannot be compared here.
**Importantly, in the case where results are only based on visual inspection, this should be stated clearly, and the authors should avoid mentioning numerical indicators, in the description of these results. For example, while it is clear that trace curve analysis (Figure 2) is based on visual inspection, the authors write “depressed women had higher mean scores of anger/irritation, anxiety/tension, and increased sensitivity”, suggesting that the results are based on numerical (as opposed to graphical) data.
Unfortunately, we cannot agree with the reviewer on this point. The sentence the reviewer quoted concerns the DIF analysis and has been described in its context. This sentence does not apply to visual inspection of the curves. The full description reads as follows: "To compare differences in item functioning between women with and without depression, the DIF function was utilized. We used the likelihood ratio (LR) c2 test as the detection criterion at the a level of 0.05, and the McFadden’s pseudo R2 as the magnitude measure. 7 items displayed depression-related DIF: 1, 2, 3, 8, 11, 12 and 14. In Figure 2, there are additional gray line curves drawn for the items which represent women with depression. On average, depressed women had higher mean scores of anger/irritation, anxiety/tension, and increased sensitivity."
**Similarly, I do not understand why statistics could not assess the differences in diagnoses proportions based on the different screening tools, across the different groups as presented in Figure 4 “Percentage of women with PMS and PMDD diagnosed by the prospective calendar of daily symptoms or retrospective PSST among women with and without depression”. Chi square tests can be used for each comparison (PSST versus Calendar in each of the 4 groups).
As much as we appreciate the reviewer’s suggestions, we cannot make the requested changes. After reconsidering the issue, we have decided to stand by the reasons we gave in our previous rebuttal. Again, we want to emphasize that Figure 4 was created to illustrate the percentage of women diagnosed with PMS/PMDD with the use of the calendar and PSST. No additional statistics can be added to the graph as it is not a simple histogram - the variables are nominal (not continuous/numerical) and the graph refers only to those who had a positive diagnosis of PMS/PMDD (without those who were not diagnosed with these syndromes). Chi-square tests can verify differences between all four groups of women and not between those who were positively diagnosed by the calendar and PSST. Moreover, it was possible for a particular participant to obtain two positive diagnosis at once (both by the PSST and the calendar). We could have divided the participants into more groups, but it would only create more confusion and still would not give us the possibility of making comparisons requested by the reviewer.
**Line 91: ”We hypothesized that depression would affect the overdiagnosis of PMS/PMDD in both prospective and retrospective methods.” Please replace “affect” by “contribute to” or “lead to”.
We thank the reviewer for pointing this out. We have corrected this statement.
**"A description of analyzes taking into account the phase of the cycle" (from the authors replies) was not found in the new version of the manuscript.
This information is in line 226.
Reviewer 2 Report
I have an additional major concern that prevents me from evaluating the adequacy of revisions to this point-- specifically, I did not realize previously that people taking oral contraceptives (who are therefore not ovulating, not experiencing menstrual cycles, and therefore not eligible to be described as "PMS/PMDD") were actually included in the IRT analyses. This is unacceptable given that oral contraceptives completely eliminate the fundamental context (the menstrual cycle) in which PMS/PMDD are defined to occur. At best, one could discuss withdrawal from exogenous hormones as an alternative kind of hormonal trigger, but the shape of such a trigger would be entirely different from a natural cycle.
Those on OCs must be removed from the sample for any analyses of symptoms across the cycle to provide useful information (since again, there is no hormone "cycle" in those who are taking oral contraceptives).
Author Response
** "I have an additional major concern that prevents me from evaluating the adequacy of revisions to this point-- specifically, I did not realize previously that people taking oral contraceptives (who are therefore not ovulating, not experiencing menstrual cycles, and therefore not eligible to be described as "PMS/PMDD") were actually included in the IRT analyses. This is unacceptable given that oral contraceptives completely eliminate the fundamental context (the menstrual cycle) in which PMS/PMDD are defined to occur. At best, one could discuss withdrawal from exogenous hormones as an alternative kind of hormonal trigger, but the shape of such a trigger would be entirely different from a natural cycle.
Those on OCs must be removed from the sample for any analyses of symptoms across the cycle to provide useful information (since again, there is no hormone "cycle" in those who are taking oral contraceptives)."
We would like to thank the reviewer for this comment. Unfortunately, we do not share the reviewer's opinion on the matter. In most stuies on PMS/PMDD there is no assumption taht taking OC equals the elimination of the menstrual cycle. Moreover, many studies show that taking OC does not reduce PMS/PMDD symptoms. One is the large population study we cited, but there are also studies by Graham et al. (1992); Backstrom et al. (1992); Sulak et al. (2000) or Joffe et al. (2003). In our study (that we referred to in the article), there were also no differences in the prevalence of PMS/PMDD between women taking OC and not taking OC. This phenomenon is best described by Coffee et. al (2006) who found that endogenous estradiol levels rise at the end of the 7-day hormone free interval, peak in the first half of the active pill cycle, and then decline during the last week of active pills. Moreover, "This decline in endogenous estradiol levels during the last week of active pills may be responsible for the estrogen-withdrawal symptoms". Thus the exclusion of OC users is neither necessary nor a standard practice in PMS/PMDD studies. The problem would arise if we only diagnosed PMS with a questionnaire. We believe that the diagnosis of PMS/PMDD by the daily calendar protected us against such misdiagnosis.
However, we do take into account that potential readers of the article may have doubts similar to the reviewer’s. Consequently, we have expanded the discussion by adding information about the Coffee et al. (2006) study describing hormonal changes in the 7-day hormone free interval period (line 437).
Round 3
Reviewer 1 Report
I included my comments in the attached document.

Reviewer 2 Report
Unfortunately, I do not agree with the authors that the ovulatory menstrual cycle is an "optional" aspect of the definition of PMDD, and I continue to have the same concerns I had before.
Here is how I defend my stance:
- While oral contraceptives may still allow some hormone flux, this will vary greatly across types, and oral contraceptives ALWAYS work in part by preventing ovulation, which is the hormonal event that triggers what we call "the menstrual cycle."
- https://journals.sagepub.com/doi/10.1177/0956797610368062
- https://www.sciencedirect.com/science/article/pii/S0015028299002058?casa_token=IxaaQGcrQRQAAAAA:1ffodQwu-73h3LXR6PIHGDP3ySzGoWRSuNUMG5sTQLxuuEwnjpESvE6IEVklb8lecbR4DG5eFA
- Even if hormone sensitivity (which may be similar in content to PMDD) occurs in response to oral contraceptives, the hormonal context would be so different (in those on vs. off OCs) as to make any comparison between trajectories inappropriate.
- PMDD is a reaction to the physiological menstrual cycle, which, as noted above, is defined by ovulation. In the DSM-5, (page 174-175), it says "Some women who present with mdoerate to severe premenstrual symptoms may be using hormonal treatments, including hormonal contraceptives. If such symptoms occur after initiation of exogenous hormone use, the symptoms may be due to the use of hormones rather than to the underlying condition of premenstrual dysphoric disorder. If the women stops hormones and the symptoms disappear, this is consistent with substance/medication-induced depressive disorder."
- Therefore, while hormone sensitive responses to OCs are possible, these do not indicate PMDD. Further, since the putative hormonal trigger in those with hormone sensitivity in reaction to the cycle vs. the pill would be completely different, this undermines the trajectory analyses presented.
- Perhaps the authors want to rename the paper/wording throughout to avoid the use of "PMDD" and instead talk about trajectories of hormone-sensitive mood symptoms. I will not accept the idea that PMDD occurs in the context of OCs, but I could accept that cyclical symptoms occur. If that change were undertaken by the authors, I would also expect some kind of direct comparison of cyclicity in the two groups.